# Deformation Behavior and Microstructural Evolution of T-Shape Upsetting Test in Ultrafine-Grained Pure Copper

**DOI:** 10.3390/ma14174869

**Published:** 2021-08-27

**Authors:** Hongpeng Jiang, Guangqiang Yan, Jianwei Li, Jie Xu, Debin Shan, Bin Guo

**Affiliations:** 1State Key Laboratory of Advanced Welding and Joining, Harbin Institute of Technology, Harbin 150001, China; jianghp1995@163.com (H.J.); yanguangqiang1@126.com (G.Y.); 13B909069@hit.edu.cn (J.L.); shandebin@hit.edu.cn (D.S.); guobin@hit.edu.cn (B.G.); 2School of Materials Science and Engineering, Harbin Institute of Technology, Harbin 150001, China

**Keywords:** T-shape upsetting test, micro-forming, equal-channel angular pressing, ultrafine grains, pure copper

## Abstract

Ultrafine-grained (UFG) materials can effectively solve the problem of size effects and improve the mechanical properties due to its ultra-high strength. This paper is dedicated to analyzing the deformation behavior and microstructural evolution of UFG pure copper based on T-shape upsetting test. Experimental results demonstrate that: the edge radius and V-groove angle have significant effects on the rib height and aspect ratio λ during T-shape upsetting; while the surface roughness has little effect on the forming load in the first stage, but in the second stage the influence becomes significant. The dynamic recrystallization temperature of UFG pure copper is between 200 °C and 250 °C.

## 1. Introduction

The sharp rise of Micro System Technology (MST) and Micro Electro Mechanical Systems (MEMS) has driven the current industrial development towards miniaturization even microminiaturization [1]. Compared with Micro-machining, Laser technology, LIGA technology and micro electro-discharge machining, metal plastic deformation has greater advantages in terms of cost control, processing efficiency and dimensional accuracy control [2,3], thus promoting the industrialization of micro-formed parts. In contrast, when using traditional plastic processing to produce micro parts, there is an obvious size effect due to the grain size being comparable to the size of the micro part [4], which would affect the final dimensional accuracy of the formed part in actual production, leading to higher costs and lower production efficiency. Studies have shown that the above-mentioned drawbacks can be significantly ameliorated when using ultrafine-grained materials [5,6], which paves the way for the development of micro-forming technology. To date, such techniques as equal-channel angular pressing (ECAP) [7,8,9], high-pressure torsion (HPT) [10] and other methods have been developed to refine the grains to micron and submicron grains of intense plastic deformation severe plastic deformation (SPD) techniques. Among them, ECAP processing is considered to be the most promising method with bulk three-dimensional isometric grains UFG material preparation.

Due to the benefit of bulk production, considerable investigations were conducted in micro forming over recent decades [11,12] to investigate the deformation behavior and friction factor of metals, mainly including ring compression, double-cup extrusion and spike forging test [13,14]. However, these tests each have its advantages and shortcomings, which can’t reflect the actual cold forging process. Zhang et al. [15] firstly proposed T-shape upsetting, a rather new method, which nearly realized the actual cold forging condition of the billet. Deng et al. [16] used the T-shape test for rectangular billets to evaluate the friction conditions of ribbed 6061 aluminum alloy parts and revealed that this method is more accurate than the ring test and double cup test. Meanwhile, the finite element method (FEM) [17,18] combined with experiments is widely used to analyze the deformation behaviors during forming. Fereshteh–Saniee et al. [19,20] conducted experimental and numerical studies on several magnesium alloys such as AZ31 and AZ80 and found that the friction sensitivity of the T-shape experiment increased with decreasing die edge radius, increasing test temperature and strain rate. FEM predictions then showed that the simulation curve matched well with the experimental load curve. Ben et al. [21,22] tested the effect of the friction factor in re-oscillation forming condition of aluminum alloy specimens using the T-shape upsetting test, and by comparing the deformed shape between the experiment and the simulation. The inverse calibration of the friction factor in the oscillation experiment was carried out. Compared with conventional extrusion experiments, the literature presented above indicates that T-shape upsetting test has the advantages of conditions similar to practical forming operations, simple die structure and easy operation, the metal is intensely deformed by both extrusion and compression simultaneously [23], whereas analysis of T-shape test in UFG materials has not yet been reported.

In plastic bulk processing, the focus of research is nothing more than microstructure and mechanical properties [24,25,26], and complex microstructures are the key factors to determine the mechanical properties of the extruded parts, so it is particularly important to study the microstructural evolution of the extrusion process. Gubicza et al. [27] studied the thermal stability of UFG pure copper prepared by different processes and found that the HPT process showed an increase in grain size along with a decrease in dislocation density, while these two changes were not significant in the specimens treated by ECAP process. By comparing the effect of heat treatment on the microstructure and thermal stability of UFG pure copper specimens with those annealed after cold rolling, Molodova et al. [28] found that the thermal stability of UFG pure copper prepared by the ECAP process was poor. In addition, Han et al. [29] studied the effect of strain rate and temperature on a unidirectional compression test of UFG Zr and revealed that deformation mechanism at higher deformation temperature and lower strain rate may be related to the dynamic recovery and dynamic recrystallization. However, the above-mentioned method is limited to traditional tension and compression deformation. Fereshte–Saniee et al. [19] showed the microstructures of the cast AZ80 alloy during T-shape deformation and found that the grains were uniformly elongated at the edge radius of the grooved die, while the grains in the middle part were less sensitive to the friction and kept its initial uneven structure.

Until now, preparation of UFG materials and characterization of mechanical properties were widely conducted, but it has not been sufficiently studied in deformation behavior combined with T-shape upsetting, especially the microstructural evolution of UFG materials at high temperature are even less. Thus, an analysis on the relationship between forming condition and deformation behavior, viewed from the microstructural evolution, would be possible to achieve a more substantial improvement to fabricate corresponding micro components with high quality.

## 2. Materials and Methods

### 2.1. Materials

A high-purity 99.99% copper bar with diameter of 2 mm was cut into short bars of 3 mm each by wire cutting as specimens and then processed by ECAP. When the material is fully lubricated, the equivalent strain of the material through a single pass is only related to the internal angle Φ and outer arc angle Ψ. In this research, the processing was conducted with die parameters Φ = 110° and Ψ = 20° as shown in Figure 1. In order to obtain UFG copper with uniform microstructure and grain size in the nanometer range, processing route Bc is utilized to extrude the raw pure copper bar up to 12 passes, more detailed descriptions of the ECAP procedure were given in earlier reports [30,31,32]. The equivalent strain can be described by Equation (1):(1)$$\varepsilon_{N}\;=\;\frac{N}{\sqrt{3}}\left\{2\cot\left(\frac{\Phi }{2}\;+\;\frac{\Psi }{2}\right)\;+\;\Psi \mathrm{cosec}\left(\frac{\Phi }{2}\;+\;\frac{\Psi }{2}\right)\right\}$$
in which $\;\varepsilon_{N}$ is the total equivalent strain during ECAP and *N* is number of passes.

### 2.2. Methods

To carry out the T-shape test, an electronic universal testing machine was used with a maximum load of 10 kN. When the specimen is pressed between die and punch, simultaneous compression occurred at both left and right sides, with extrusion in the middle part taking place, so that the specimen changes to “T-shape” from cylindrical. The forming load and the height of the extruded middle part (rib height) are subject to various factors such as grain size, die parameters, testing temperature and lubrication conditions [23]. In this research, the width of the V-groove is fixed at 1 mm, the roughness of die is *Ra* = 0.2, *Ra* = 0.8, *Ra* = 1.6 and *Ra* = 3.2, the V-groove angle is 15°, 20°, 30° and 45°, and the edge radius is 0.1 mm,0.2 mm,0.3 mm,0.4 mm and 0.5 mm. The load was set from 2500 N to 4000 N with an interval of 500 N. As shown in Figure 2, after upsetting, the height(H) and width(D) of the extruded part were measured by vernier calipers, and the changes of aspect ratio λ(H/D) were further analyzed to evaluate the influence of the above-mentioned parameters as well as metal flow during T-shape upsetting test.

The microstructural evolution of all T-shape test samples was observed by Quanta 200FEG field emission scanning electron microscope (FESEM). Before EBSD measurement, the cross section of each observation region was ground by abrasive papers grit 200-1000, then the samples were electropolished until mirror-like surfaces using a solution with ratio C_2_H_5_OH:H_3_PO_4_:H_2_O = 1:1:2, which was maintained at 0.1 A, 6 V for 3 min. In the subsequent data processing, the scanning results are first denoised, and the data with confidence index (CI) greater than 0.35 is analyzed to study the influence of load and testing temperature on grain orientation, grain size, and grain boundary distribution.

## 3. Results

### 3.1. FEM Analysis

Deform-3D finite element software was used to simulate the deformation behavior in the T-shape upsetting process. The conditions were set with consistence with the actual experimental parameters. In order to investigate the forming process, three parameters were selected as material flow rate, equivalent stress, and equivalent strain. From Figure 3a, it can be seen that at the beginning of the T-shape upsetting, the metal flow rate into the cavity is greater than that to the lateral, while the opposite is the case in the late stage. Figure 3b,c illustrate the contour of the equivalent stress and equivalent strain distribution during deformation, respectively. In Figure 3b, as the deformation processing, the contact area as well as the volume of the stress zone increases. Figure 3c shows that the deformation mainly occurs near the edge of the die, the maximum equivalent strain increases at first and then decreases and reaches the maximum value when the flow rate to lateral identical to that into the groove. 

### 3.2. Effect of Surface Roughness

To investigate the effect of surface roughness during deformation of UFG copper, four surface roughness with *Ra* = 0.2, *Ra* = 0.8, *Ra* = 1.6 and *Ra* = 3.2 are chosen. Stroke-Load curves for coarse-grained (CG) copper and ultrafine-grained (UFG) copper with variation of *Ra* are shown in Figure 4a,b. As *Ra* increases from 0.2 to 3.2, the rib height increases by 0.11 mm and 0.1 mm for CG and UFG pure copper, respectively. Namely, with the increase of the friction factor, the load gradually increases, which went well with the research of Zhang [13]. In the first stage, the space inside the groove is spacious, and the deformation resistance generated mainly comes from the plastic flow of UFG copper. Therefore, in this stage the friction factor is less susceptive with the load. While in the second stage, the groove has been mostly filled with metal, and the contact area between the specimen and die increases rapidly when the frictional resistance is in prominent. Due to no constraint between the top surface of the die, the flow to lateral is accelerated at this time, coming with the rapidly increasing frictional force between the specimen and the top surface of the die, which mainly hinders the flow to lateral. While in the groove the space is gradually reduced, and the compression force between the groove and the specimen increases, the frictional resistance on the interface hinders the material from flowing to the inside of the groove. Besides, as the frictional force increases, the vertical component also increases. Hence, the larger the surface roughness of the die, the larger the friction factor between the specimen and the die, and in the late stage of deformation, the frictional resistance to be overcome increases as well as the forming load.

Figure 5 shows the variation curves of λ with the surface roughness in CG copper and UFG copper, respectively. The discrete points are the experiment data, and the curve is the result of linear fitting by Origin. As the surface roughness increases, the aspect ratio λ of the specimen keep decreasing with a linear correlation. When the roughness increased from 0.2 to 3.2, the aspect ratio λ decreased 3.9% and 6.7% for CG and UFG pure copper, respectively. We can see the CG pure copper curve is above the UFG pure copper curve, which can be concluded that during the preparation of UFG copper, plastic deformation occurs more dramatically than CG copper, so the deformation resistance of UFG is higher, and therefore easier to reach the limited load. When the material flows inside the die groove, a rate gradient is formed at its bottom along the width direction due to friction at the interface with the die, with the maximum flow rate at the middle position and the minimum flow rate at the interface. Accordingly, as the frictional force increases, the flow rate at the bottom of the specimen decreases. Moreover, when the forming load remains, the larger the surface roughness is, the more difficult the metal flow into the groove. In summary, the increase of the friction factor hinders the metal pushed into the V-groove and the material tends to expand to lateral, which is consistent with the simulation results in Section 3.1.

### 3.3. Effect of Edge Radius and V-Groove Angle

In this subsection, two crucial die parameters, edge radius and V-groove angle, are considered, with limited load of 4000 N (Figure 6). The results show that edge radius and V-groove angle imply the uniform impact on deformation process, which present a linear negative correlation with λ. This result can be explained that compared with Ra, both large edge radius and V-groove angle contribute to a more spacious space for metal flow. Especially with the increase of V-groove angle, the vertical component of friction generated on the V-groove becomes smaller, which induce the decreasing of rib height sensitivity.

### 3.4. Effect of Testing Temperature

Except for the influence of die parameters, the testing temperature should as well be considered. Therefore, T-shape test ranges from ambient temperature to 400 °C using a die with V-groove angle of *β* = 20° and *Ra* = 1.6 were conducted. In order to ensure temperature stability during deformation, a temperature control box is applied for heating, and thermocouples are used for real-time monitoring. Considering that the increasing temperature would soften the material, the upsetting rate was fixed at 0.2 mm/s and the maximum load was 3500 N.

Figure 7a,b present the Stroke-Load curves under seven different testing temperature, respectively. Inspection of CG and UFG curves show that the required load decreases at the same displacement at elevated temperature; that is, the increasing temperature can effectively reduce the forming load during T-shape upsetting, especially for UFG copper in the first stage the effect is more significant. The load is less sensitive when the temperature is lower than 200 °C, and becomes sensitive when the temperature exceeds 250 °C. However, when comes to the second stage, the slope of the curve gradually increases with the elevated temperature. One reason for this phenomenon is that in the second stage, the interface between the specimen and die increases rapidly as well as friction resistance. On the other hand, the material gradually softens with the elevated temperature also contributes to a greater real contact area at the interface, also resulting in the larger friction resistance. In contrast, the CG copper has a less temperature sensitivity with inconspicuous changes of Stroke-Load curves, similar behavior was observed by Tao et al. [33] for CG copper under uniaxial compression at elevated temperature, which indicates that the grain refinement augments the temperature sensitivity.

Figure 8 illustrates the variation curves of λ with the testing temperature of CG and UFG copper, respectively. Under temperature lower than 200 °C, dynamic softening is dominated by dynamic recovery due to the driving force to promote recrystallization nucleation and grain growth is small, resulting in a lower rib height. In contrast, the part between the punch and die expanded wider to lateral due to the material softening, which may be the major contribution to the decrease of the aspect ratio λ. However, when the testing temperature exceeds 200 °C, the increasing temperature provides a superior driving force for dynamic recrystallization, the larger recrystallized grains and higher material flow contribute to the higher rib height as well as λ. For CG pure copper, λ increases linearly with the testing temperature, and reaches its maximum value when the temperature reaches 300 °C. When the testing temperature exceeds 400 °C, λ starts to decrease.

### 3.5. Microhardness

To analyze the differed plastic deformation mechanisms for UFG pure Cu at elevated temperature during the T-shape upsetting process, the microhardness distribution diagrams are obtained and shown in Figure 9 and Figure 10, respectively. Due to the severe plastic deformation during ECAP, hence generating numerous high-density dislocations in the grain boundaries, which introduces the higher microhardness of the initial UFG copper than CG copper. After upsetting at ambient temperature, the UFG copper undergoes work hardening, and the microhardness increases significantly. It was apparent that dynamic recovery and dynamic recrystallization occurs to balance the dislocation generation at elevated temperature condition, thereby leading to a reduction of microhardness in UFG copper. Moreover, when the testing temperature exceeds 250 °C, the microhardness after upsetting drops below the initial value of the sample. For CG copper, the positive work hardening still exists at 400 °C, since the enhanced dynamic recovery of dislocations under 400 °C was not enough to offset the work hardening, which generally occurred in face-centered cubic CG metals.

### 3.6. Microstructural Evolution after T-Shape Upsetting

Microstructural evolution plays a key role in evaluating the mechanical properties of UFG copper. Four prominent positions of the specimen were observed by electron back scatter diffraction (EBSD) and analyzed by TSL orientation imaging microscopy (OIM) after T-shape upsetting of UFG copper. 

The experiment was conducted at condition with V-groove angle β = 20°, edge radius R = 0.2, roughness Ra = 1.6 and load 3500 N. As shown in Figure 11, where Figure 11a–d represent the region A, B, C and D after deformation, respectively. In these and all subsequent OIM images, the grain boundaries are denoted either by green lines corresponding to low-angle grain boundaries (LAGBs) between 2–15° or black lines corresponding to from high-angle grain boundaries (HAGBs) above 15°.When the T-shape upsetting occurs, region A firstly contacts with the punch. The specimen expands rapidly, and the contact area with the punch gradually becomes larger. In the subsequent deformation process, due to the restriction by the surrounding metal, the grain orientations and the microstructure changes are unobvious in region A. In the second stage of deformation, the metal flow rate into the groove is slow, but the expansion rate to the surrounding area increases, which resulting in the flattening grains, grain orientation tends to be identical to the deformation direction in region B. The most severe deformation occurred at the edge corner of the die (region C) and the grains are gradually elongated along the deformation direction from the initial equiaxed shape to strip-like shape. In addition, both extrusion and friction in the corner introduce the different degrees of grain deformation, in which the grain elongation is most serious in the part closest to the die, and the elongation phenomenon is unobvious away from the die. It is readily apparent that grain orientation in region D is consistent with the flow direction in vertical, and the grain size remains after upsetting test. Thus, it is believed that grain rotation is the main mechanism during the deformation process in region D.

### 3.7. Effect of Temperature on Microstructural Evolution

Specimens in 3.4 were chosen to characterize the effect of testing temperature on T-shape upsetting of UFG copper. The microstructural evolution in region C is summarized in Figure 12. Inspection of Figure 12a,b show that the ultrafine grains can keep equiaxed and uniformly distributed under 200 °C, which was mainly influenced by drastic plastic deformation. With the temperature increasing to above 250 °C, the grain grows rapidly and becomes coarser, but the grain elongation appears indistinct at the edge corner. Previous researchers have revealed that the recrystallization temperature for industrial CG pure copper is between 300 °C and 350 °C [34]. However, for the UFG copper prepared by the ECAP method, the material has undergone intense plastic deformation, high density dislocations inside the grains, high internal stresses, and uneven energy distribution, which enhance the conditions for its annealing and recrystallization nucleation. Besides, the load applied by the punch in upsetting process also provides energy for the nucleation of UFG pure copper. Therefore, even at a temperature lower than the recrystallization temperature of CG copper, the internal energy of the UFG copper grains will be slowly emitted. Additionally, for this experiment, when the testing temperature is low, the time required for air cooling to room temperature after the experiment is short. Therefore, when the testing temperature is lower than 200 °C, the grain size of UFG pure copper grows slowly, but the easier grain elongation was observed, which is attributed to the slight softening of materials. It should be noted that the recrystallization driving force is larger when testing temperature exceeded 250 °C, and the die as well as specimen are still at a higher temperature after the test, coming with the extended time required for cooling to ambient temperature, which indicates that the grain size grows faster. However, attributed to the short time during the deformation process, the grain size becomes non-uniform after recrystallization. Moreover, the grain boundaries are re-integrated and rearranged during the grain growth process, so the grain elongation phenomenon cannot be observed.

Figure 13a shows the detailed average grain size versus the corresponding testing temperature after T-shape upsetting of UFG pure copper. The average grain size is directly calculated by the EBSD software OIM. When the testing temperature is 200 °C, the average size of the grains only increases by 0.2 μm contrast with ambient condition; when the temperature exceeds 250 °C, the grain size increases almost linearly with the testing temperature. When testing temperature further reaches 400 °C, the average size of the grains grows from 0.44 μm to 3.40 μm. However, the grain size of conventional CG copper is less sensitive to temperature until 400 °C, which may account for the inconspicuous curve changes in Figure 7a. Hall–Petch relationship illustrates that the yield strength and microhardness of the material increase linearly with the decrease of average grain size to the power of 1/2 [35]. Analyzing Figure 13b indicates that the Hall–Petch relationship is still applicable for UFG materials.

Figure 14 shows the corresponding grain distributions of UFG copper under different testing temperature and Figure 15 illustrates the specific histogram statistics. It is apparent that the ratio of LAGBs gradually decreases with the elevated temperature, although misorientation angle lower than 5° accounted for more than 40% still at 150 °C. While the fraction of HAGBs appears larger with the rising temperature, when the temperature increases to 300 °C, the proportion of HAGBs exceeds 85%. Specifically, the fraction of twin boundaries increases significantly to 50% under 300 °C [36]. In-depth analyses show that dislocation caused by plastic deformation favors LAGBs turning into HAGBs. On the other hand, the increasing temperature accelerated dislocation movement degree, inducing more evident recrystallization during this period. Nonetheless, accompanied with testing, temperature further reaches 400 °C, the ratio of twin boundaries decreases, and abundant LAGBs are generated at the HAGBs. The reason for this is that under high temperature (>400 °C) the rising atomic activity accelerates mobility, causing a low probability of stacking faults, which may be the major contribution to the decreasing formation of annealing twins and twin boundaries.

The kernel average misorientation (KAM) is an effective way to qualitatively analyze plastic deformation degree within a metal, which is related to the dislocation density. [37]. Figure 16 shows the distribution of KAM inside UFG copper specimens in different testing temperatures and Figure 17a shows the quantitative analysis. Figure 17b demonstrates the corresponding microhardness in edge corner. Inspection of Figure 16 shows that the peak of the KAM increases with the elevated temperature under 200 °C after upsetting. At this time, the larger KAM value is mainly distributed in the areas where the grains are coarsened, especially the areas where the grains grow rapidly, indicating that the deformation at this temperature is dominated by dislocation movement. While the peak of KAM curve becomes wider and lower over 200 °C, the deformation degree increases with the elevated temperature, attributed to enhanced ability of dislocation movement, and some of the dislocations are offset, which triggered a strong dynamic recovery and recrystallization. Meanwhile, the microhardness decreases slowly from Figure 16b, but is still higher than the microhardness of undeformed UFG copper. When the testing temperature is higher than 200 °C, intragranular dislocation movement account for the drastically reduction of dislocation density, which triggered the decrease of average value of KAM. In addition, work hardening is severely weakened when the testing temperature is higher than 250 °C, the microhardness at the edge corner is significantly reduced to lower than the value of undeformed UFG copper. It can be concluded that the recrystallization temperature of UFG copper is between 200 °C and 250 °C.

## 4. Conclusions

(1)The deformation process can be divided into two stages according to the law of material flow: In the first stage, the flow rate into the groove is greater than the flow rate to lateral, while in the second stage we have the opposite.(2)The UFG copper has a strong temperature dependence, high V-groove angle and edge radius sensitivity during T-shape upsetting, while surface roughness has little effect on the change of rib height and the aspect ratio λ.(3)The most severe deformation occurred near the corner of the die, which is related with the friction and extrusion in the interface, and the initial equiaxed grains gradually evolve to strip-like grains along the deformation direction; while at the bottom part, instead of elongation, grain rotation is the main deformation mechanism during the upsetting process.(4)The grain growth has an evident discrepancy at elevated temperature, for which the dynamic recovery seems to weaken the work hardening under 200 °C, while the dynamic recrystallization gradually dominates at the deformation temperature higher than 250 °C with a rising fraction of twin boundaries.

## Figures and Tables

**Figure 1 materials-14-04869-f001:**
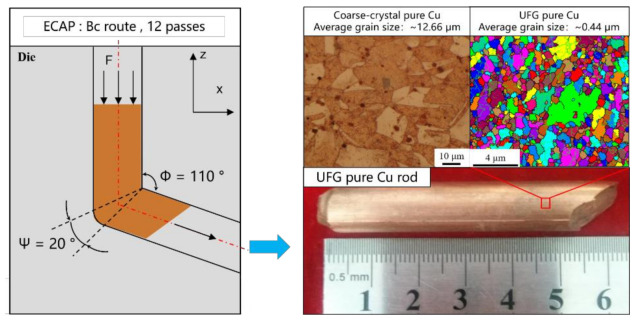
Preparation of UFG pure Cu specimen by ECAP.

**Figure 2 materials-14-04869-f002:**
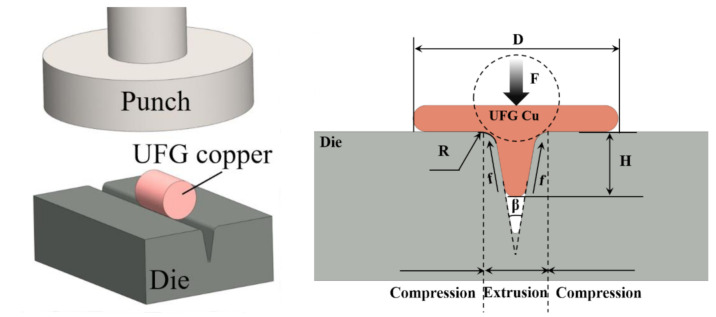
Schematic diagram of T-shape upsetting.

**Figure 3 materials-14-04869-f003:**
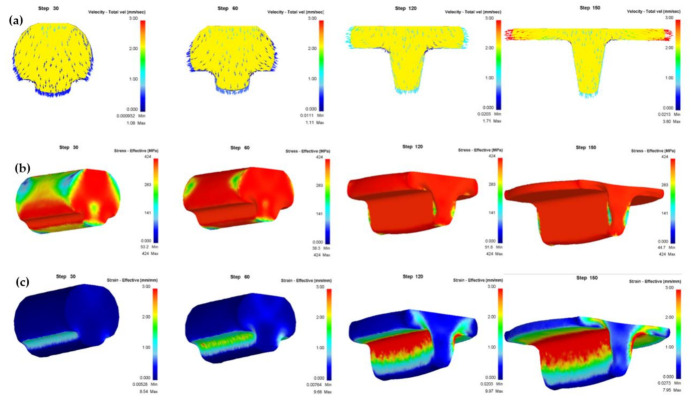
Analysis of T-shape upsetting process in UFG pure copper: (**a**) material flow rate distribution; (**b**) equivalent stress distribution; (**c**) equivalent strain distribution.

**Figure 4 materials-14-04869-f004:**
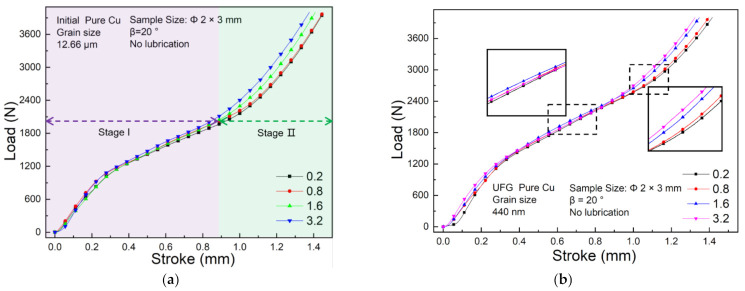
Stroke-Load curve under different surface roughness: (**a**) CG Cu; (**b**) UFG Cu.

**Figure 5 materials-14-04869-f005:**
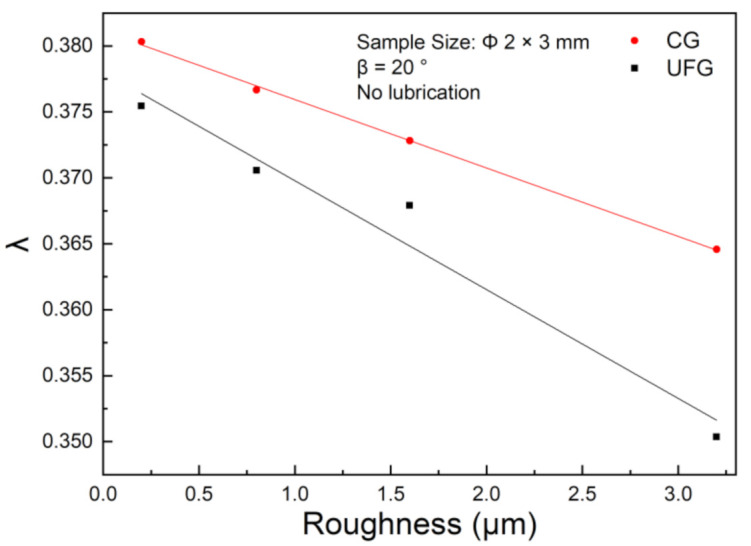
Variation curve of the aspect ratio λ of UFG and CG pure copper with the surface roughness.

**Figure 6 materials-14-04869-f006:**
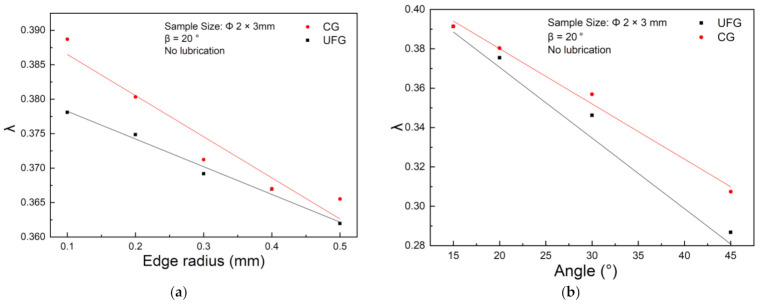
Variation curve of the aspect ratio λ of UFG and CG pure copper: (**a**) edge radius, (**b**) V-groove angle.

**Figure 7 materials-14-04869-f007:**
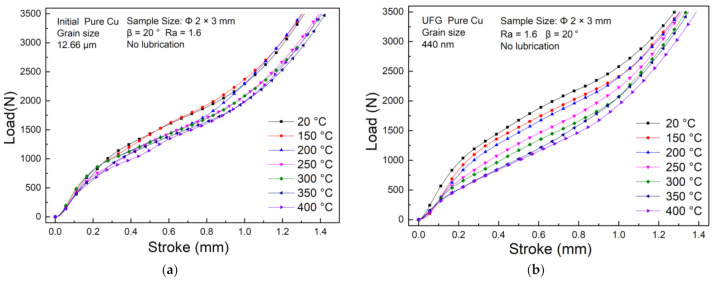
Stroke-Load curve under different testing temperature: (**a**) CG Cu; (**b**) UFG Cu.

**Figure 8 materials-14-04869-f008:**
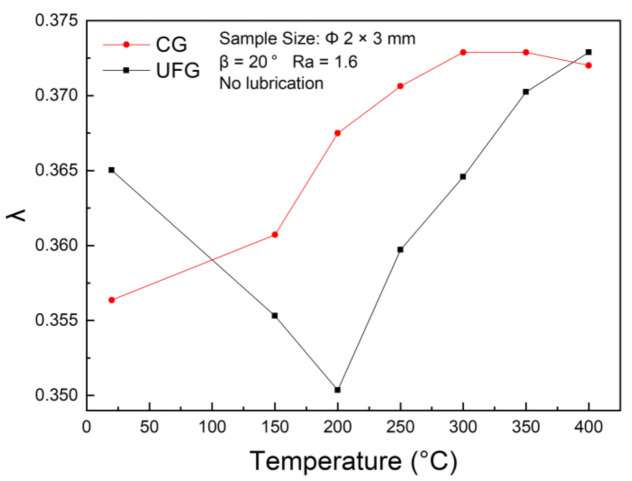
Variation curve of the aspect ratio λ of UFG and CG pure copper with testing temperature.

**Figure 9 materials-14-04869-f009:**
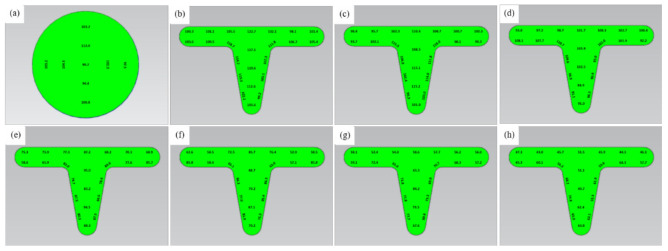
Microhardness distribution (HV) of UFG copper under different testing temperature: (**a**) original specimen, (**b**) ambient temperature, (**c**) 150 °C, (**d**) 200 °C, (**e**) 250 °C, (**f**) 300 °C, (**g**) 350 °C, (**h**) 400 °C.

**Figure 10 materials-14-04869-f010:**
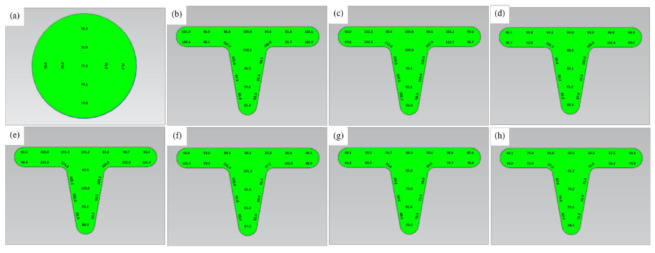
Microhardness distribution (HV) of CG copper under different testing temperature: (**a**) original specimen, (**b**) ambient temperature, (**c**) 150 °C, (**d**) 200 °C, (**e**) 250 °C, (**f**) 300 °C, (**g**) 350 °C, (**h**) 400 °C.

**Figure 11 materials-14-04869-f011:**
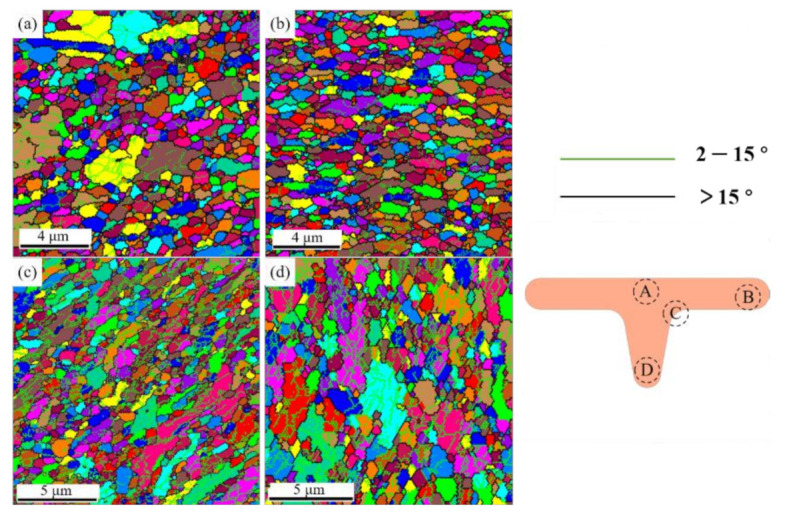
Microstructural evolution in primary regions of UFG pure copper after T-shape upsetting: (**a**) region A, (**b**) region B, (**c**) region C, (**d**) region D.

**Figure 12 materials-14-04869-f012:**
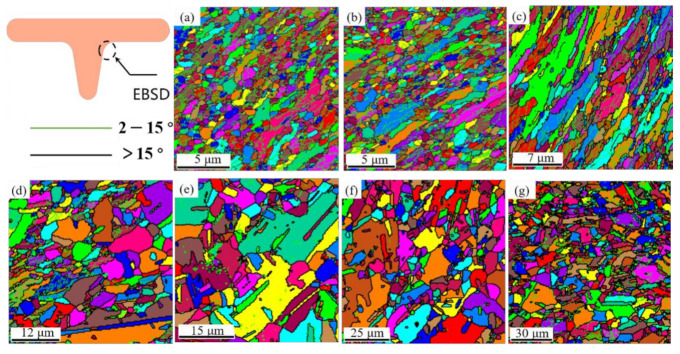
Microstructure after T-shape upsetting of UFG pure copper at different temperatures: (**a**) room temperature; (**b**) 150 °C; (**c**) 200 °C; (**d**) 250 °C; (**e**) 300 °C; (**f**) 350 °C; (**g**) 400 °C.

**Figure 13 materials-14-04869-f013:**
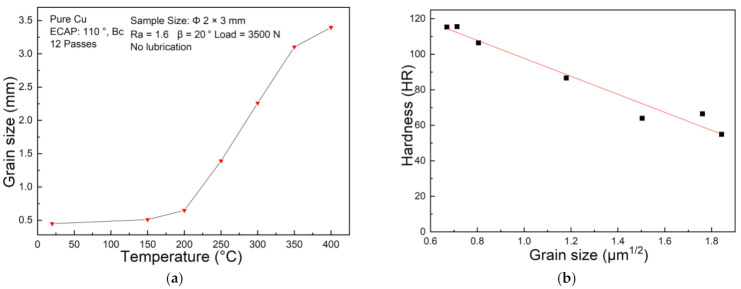
(**a**) Average grain size at different deformation temperature in UFG copper after T-shape upsetting; (**b**) Microhardness versus average grain size.

**Figure 14 materials-14-04869-f014:**
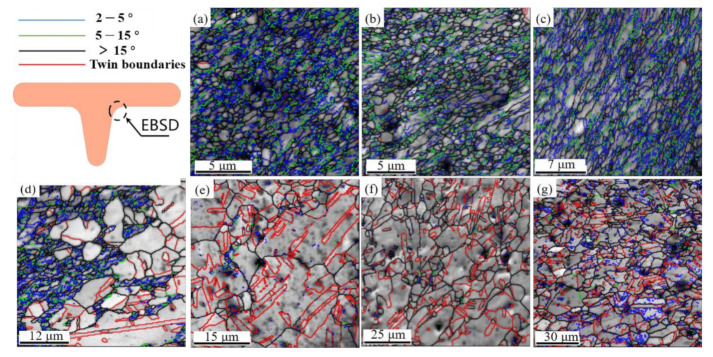
Grain boundary distribution of UFG pure copper under different testing temperature conditions: (**a**) room temperature; (**b**) 150 °C; (**c**) 200 °C; (**d**) 250 °C; (**e**) 300 °C; (**f**) 350 °C; (**g**) 400 °C.

**Figure 15 materials-14-04869-f015:**
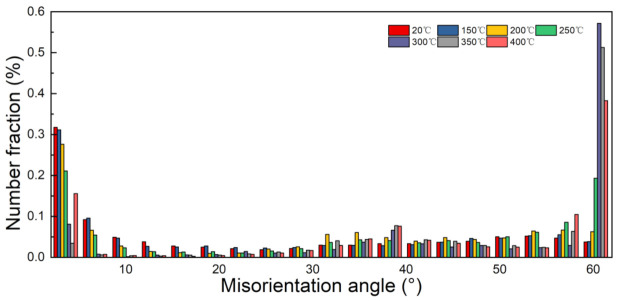
Histogram of the misorientation distribution of UFG pure copper under different testing temperature.

**Figure 16 materials-14-04869-f016:**
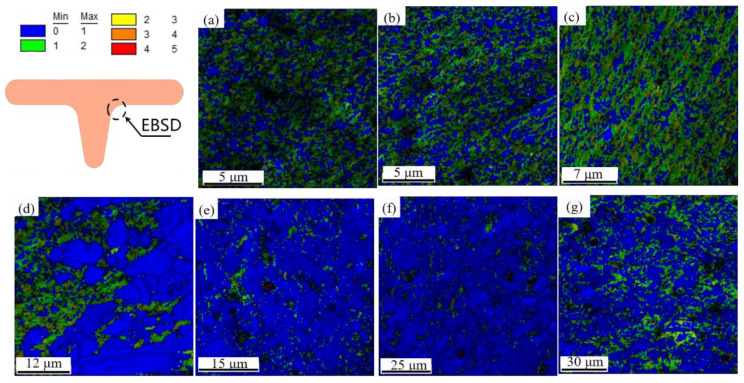
KAM distribution of UFG pure copper under different testing temperature conditions: (**a**) room temperature; (**b**) 150 °C; (**c**) 200 °C; (**d**) 250 °C; (**e**) 300 °C; (**f**) 350 °C; (**g**) 400 °C.

**Figure 17 materials-14-04869-f017:**
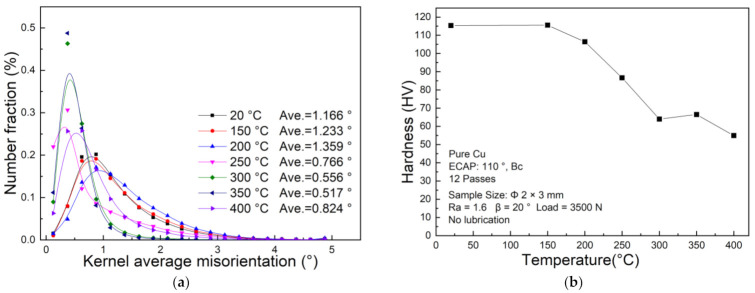
(**a**) The quantitative analysis of the KAM distribution inside the UFG pure copper specimens with different testing temperatures, (**b**) The microhardness distribution at edge corner after T-shape upsetting at different temperatures.

## Data Availability

The data presented in this study are available on request to the corresponding author.

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
