# Peer review of "Deformation Behavior and Microstructural Evolution of T-Shape Upsetting Test in Ultrafine-Grained Pure Copper"

_materials, 2021, doi:10.3390/ma14174869_

Round 1
Reviewer 1 Report
Ultrafine-grained (UFG) pure copper 99,9% developed through ECAP (Equal-channel angular pressing) method was used for studying the deformation behaviour and microstructural evolution with a T-shape micro-upsetting.
This T-setup has the advantages of a simple die structure and easy operation and, the material is subjected both to extrusion and compression in compare with the traditional compression test. Also, the same set of die can be used for specimens with different sizes.
In this study, the T-shaped experiment was used to analyse the influence of the surface roughness of the die, the die parameters on the rib hight under temperature, the deformation of the metal and the microstructure evolution.
This article is well written and organized, and the results are clearly detailed, and the discussion is also adequate.
My recommendation is: accept for publication.
Author Response
Thanks for your kind work and approval. Besides,we have improved the written of paper for better understanding.
Reviewer 2 Report
Dear authors,
thank you for the interesting material. In general, the article is worthy of being published in the Materials, but it is necessary to pay attention to the following points.
1. Lines 101 and 105. The same designation is used to indicate the diameter of the workpiece and the angle of the die.
2. Lines 108 and 111. Most likely the word "strain" is missing. What is eN? Equivalent strain?
3. Line 116. The abbreviation for kilonewtons is kN, not KN.
4. Sections 3.4 and 3.6. It is not correct to evaluate the effect of the process temperature without taking into account the extrusion rate (strain rate). The recrystallization process is directly related to both temperature and strain rate. In addition, it is not described how the temperature of the billet was controlled during the upsetting process? Was the die and punch warmed up? Given the small size of the billet, upon contact with cold die/punch, its temperature could significantly decrease.
5. Section 3.5. It is necessary to give for which process parameters (load, angle, roughness, radius) the microstructures are shown in Fig. 9. How do these parameters affect the microstructure?
Reviewer 3 Report
This work focuses on the preparation and testing of UFG samples for forming. The principal means of demonstrating the effect is by comparison with CG samples. The latter part of the manuscript, however, contains imagery and analysis of only the UFG samples. No conclusions can be drawn regarding the effect of grain refinement.
To reach any conclusion from the results that are provided, basic experiments into plastic properties (i.e., compression or tension tests) as well as friction (e.g., ring tests) need to be performed. The ECAP processing begs the question of what the refinement did to the properties.
Upsetting tests have been done for decades. The authors need to consult the works of Altan, Kobayashi, Avitzur, Mielke, and others.
Figure 7 is very interesting and needs further explanation. To make them easier to interpret, the scaling of the axes of the two plots should be the same. It appears the grain refinement did not lead to larger forces for a given stroke when tested at room temperature. This is surprising and needs explanation. At elevated temperature, the authors indicate there was time for the UFG grains to change before, during, and after each test. These experiments therefore do not seem to have had sufficient control.
More detailed comments:
Mention is made of both friction coefficient and friction factor. These are not the same. Also, the relation among surface roughness and friction factor needs to be established.
Reference is made to "friction force" and "deformation force" apparently in a qualitative speculative sense. The magnitudes of these would need to be estimated by also performing highly-lubricated tests.
Line 221. The term "viscosity" applies to fluid behavior. Further the phrase the "viscosity between the material and the die increases" only begins to make sense if a liquid lubricant is present. Yet, the manuscript is replete with notations that lubricant was not used.
Reviewer 4 Report
In the present manuscript, the effect of die surface roughness on the microstructure evolution of the ultrafine grain structure Copper. The topic is interesting and important for forming Cu parts. However, the manuscript lacks novelty and needs to clarify.
The introduction needs to be revised. There are many interesting new articles in the literature regarding this topic. So, please update the literature survey to find the literature gap.
Please add detail experimental; especially for the EBSD sample prep, data acquisition, and analysis.
The presented results are just reported. Please add a detailed discussion of these results.
The conclusion does not match with the content of the achievement or the content of the results.
References need to be up to date.
Reviewer 5 Report
This paper is dedicated to analyzing the deformation behavior and microstructural evolution of UFG pure copper in a T-shaped open upsetting dies. The context of the work is the forming of small metal parts for which billets with typical grain sizes would be too large for high quality results. The authors appear to intend to show the effects of grain size by juxtaposition against coarse-grain results under the same forming conditions. The independent variables include die geometry, temperature, and surface roughness.
This sort of upsetting test predates 2009 by decades. The authors need to consult the works of Altan (e.g., Forging Equipment, Materials and Practices and the references therein) as well as Kobayashi, Avitzur, Mielnik and others.
The most fascinating result is Figure 7. A sound interpretation can't be made, however, without performing additional straightforward experiments such as ring tests (to unambiguously determine the effects of surface roughness and temperature on friction factor among CG and UFG) and tensile or compressive tests (to determine CG and UFG plastic properties as functions of temperature). The ECAP processing begs the question of what it did to the mechanical properties. It would seem the grain refinement and insertion of defects from ECAP would have led to far more resistance to deformation at 20 C. However, this was not the case. Because the load scales differ among figures 7a and 7b, it is difficult to clearly identify where the quantitative differences lie. It is great to find numerous microstructure EBSD photos of UFG specimens and analyses of them. However, one can only speculate on the effect of grain size, recovery, and recrystallization if microstructure photos and analyses from CG samples are not provided.
Regarding the conclusions:
1) Unclear. The authors mention the "deformation force" and "friction force" in the manuscript. However, they are not separately measured. Further, the effects of geometry and friction on upsetting have been explored for decades.
2) While dynamic recovery may be the only active mechanism below 200 C, it is unclear why should it be the case that lambda decreased with increasing temperature up to 200 C. Was frictional resistance also varying? Was dynamic recovery non-uniform?
3) If this is in reference to location "C" it is difficult to know the flow direction because it is so close to the neutral point. If this conclusion also derives from Figure 10, grains do not appear elongated in some cases. Do the authors mean to say grains were elongated for cases where the temperature was below the putative recrystallization temperature?
More specific details:
In some instances, the authors refer to friction factor whereas in others they refer to friction coefficient. Please note these are not synonymous.
The "deformation force" (presumably the portion of the total force required to cause further plastic flow) and the "friction force" are mentioned numerous times. These seem to be offered qualitatively and in a speculative context. To have a good idea of the relative sizes of each, zero-friction (highly lubricated) cases would have to be run. I don't find evidence of this.
Fig 2: Was the width of the base of the rib the same in each case?
Fig. 3 The independent variables are not varied in a methodical way among the curves. The message of the figure is therefore unclear.
Line 152. Dependence of friction factor or friction coefficient on surface roughness needs to be established by a separate experiment (and not presumed).
155-156. Unclear, what is meant by "Therefore, in this stage the friction factor is less susceptive with the deformation force." ?
179-184. Speculative interpretation.
Line 219. To what is "this phenomenon" referring?
Lines 220-222 does not make sense. The term "viscosity" applies to fluid behavior. Further the phrase the "viscosity between the material and the die increases" only begins to make sense if a liquid lubricant is present. Yet, the manuscript is replete with notations that lubricant was not used.
Fig. 10. It is unclear whether there is any trend displayed in terms of grain size nor elongation nor grain growth. At the least, the scaling needs to be the same among them.
Line 294-297. The authors state the time at elevated temperature is not in good control. Therefore, it can't be determined the grain growth due to preheating, during forming, and cooling. It would seem the latter could have been avoided by quenching.
Line 305. This paragraph needs a topic sentence.
Line 305. From where in the specimen are the misorientation angles recorded?
Fig. 11. The bars in the chart are so narrow their colors can barely be seen.
Lines 333-336. Recovery and recrystallization are mentioned in the same sentence as an interpretation of the outcome. The authors need to clearly sort these from each other. The sentence almost reads as if the two phenomena are being treated synonymously.
Reviewer 6 Report
The authors provide a paper dealing with deformation behavior and microstructural evolution in ultrafine-grained pure copper. The paper can be of interest for Materials, but MAJOR revisions are requested.
- The way the paper is written must be improved. Special care should be taken to the English as well as to provide a good scientific way to present the data. A revision made by a native speaker can be very useful.
- The title must be improved. It does not have to contain “T-shape” and “Micro-upsetting” and must be more comprehensive. More detail can be provided in the abstract which also need to be clearer (see my point 1).
- In Figure 3 it is not clear what is lambda and how it is defined. This quantity can be reported in the schematic of Figs. 1 or 2.
- I think the authors should comment the mechanical properties of the materials at the microscale commenting on existing in-situ techniques which manage to extract the fracture toughness at the micro-scale in situ such as in doi.org/10.1016/j.matdes.2019.107762 or tensile test doi.org/10.1016/j.actamat.2017.03.072 and many other paper. This will be of a capital importance for the present paper aiming to improve the mechanical properties based on the ultrafine grain refinement of Cu. More in general, a comment on the expected mechanical properties at the micro-scale based on the previously mentioned techniques will be very helpful.
- The authors must provide some XRD before and after the deformation to identify the change in the atomic arrangement and evaluate the microstructural evolution i.e. change in the average crystallite domain (Sherrer equation).
Round 2
Reviewer 3 Report
The abstract states the manuscript is dedicated to analyzing the microstructural evolution of UFG copper based on a T-shaped upsetting test.
For most of the manuscript, the material in the CG state is the control. In their subsequent response regarding such a comparison, the authors make reference to their previously-published work. In these, however, the material tested was high purity aluminum. If the authors mean to state such results have already been published, is the current work is novel? However, it is more credible that the investigation presented here was performed because there are differences in the thermal, defect diffusion, and other physical processes in copper. Therefore, EBSD or some other quantitative record of grain sizing from the CG samples needs to be provided to support the manuscript's purpose implied in the abstract.
Again, an objective relation needs to be made among surface roughness and friction factor. While surface roughness is treated as an independent variable, interpretations are made based on arguments of friction. Owing to the complexities of the surface interaction, the relation can't be made via simulations. Further, the Figure 1 supplied by the authors shows the effect of roughness on friction factor is slight up to R_a = 1.6. In dry ring tests and constant surface roughness, friction factor can vary dramatically from sample to sample as well as with the reduction in height. Here is the abstract to reference 13 in the current manuscript:
Lubrication and friction conditions vary with deformation during metal forming processes. Significant macro-variations can be observed when a threshold of deformation is reached. This study shows that during the cold compression processing of #45 (AISI 1045) steel rings, the magnitude of friction and surface roughness (Ra) changes significantly upon reaching a 45% reduction in ring height. For example, the Ra of compressed ring specimens increased by approximately 55% immediately before and after reaching this threshold, compared to an 18% or 25%variation over a 35%−45% or a 45%−55% reduction in height, respectively. The ring compression test conducted by this study indicates that the Coulomb friction coefficient μ and Tresca friction factor m are 0.105 and 0.22, respectively, when the reduction in height is less than 45%; and 0.11 and 0.24, respectively, when the reduction in height is greater than 45%.
Again, material constitutive models (informed by physical experiments) need to be provided in order to interpret the results. Finite-element-based results of velocity and strain distribution are provided for one UFG case are provided. No results are provided for comparison against CG cases. Also, the underlying plastic flow constitutive model is not provided.
Figure 7 continues to show fascinating results which require more complete explanation. While new text has been added, it is unclear when and where the narrative refers to the CG case or the UFG case. There is a distinctly different dependence on temperature among CG and UFG. If the frictional conditions are ostensibly the same, it would seem the understanding of the differences has to come from comparison among the microstructures. Can the new narrative provided in 3.6 be augmented to also address what occurs during deformation in CG cases and therefore provide a basis for understanding Figure 7?
If the stated references were consulted then the need to be cited.
The conclusions principally provide observations. Instead, they should provide a more generalizable message about the core investigation (the effect of grain size on microstructure and processing thermomechanics).
While the results of much work are provided here, the story lacks comprehensiveness in terms of the both the microstructures and the thermo-mechanics.
Reviewer 4 Report
Can be accepted in the present form!!
Author Response
Thanks for your kind work and approval.Our manuscript has been reviewed by a native English speaker and revised to improve readability
Reviewer 5 Report
I believe that the publication has gained significantly after taking into account the comments of reviewers, there is already approval on my part. The linguistic evaluation is not undertaken, although the article is understandable to me.
best regards
Author Response

(The authors gave the same response as above.)

Reviewer 6 Report
-
Author Response

(The authors gave the same response as above.)
